# Towards Test-Time Refusals via Concept Negation

**Peiran Dong**[1]    **Song Guo**[2]*    **Junxiao Wang**[3]*    **Bingjie Wang**[1]
**Jiewei Zhang**[1]    **Ziming Liu**[1]

[1]Hong Kong Polytechnic University    [2]Hong Kong University of Science and Technology
[3]King Abdullah University of Science and Technology & SDAIA-KAUST AI
{peiran.dong,bingjie.wang,jiewei.zhang,ziming.liu}@connect.polyu.hk
songguo@cse.ust.hk
junxiao.wang@kaust.edu.sa

## Abstract

Generative models produce unbounded outputs, necessitating the use of refusal techniques to confine their output space. Employing generative refusals is crucial in upholding the ethical and copyright integrity of synthesized content, particularly when working with widely adopted diffusion models. "Concept negation" presents a promising paradigm to achieve generative refusals, as it effectively defines and governs the model's output space based on concepts, utilizing natural language interfaces that are readily comprehensible to humans. However, despite the valuable contributions of prior research to the field of concept negation, it still suffers from significant limitations. The existing concept negation methods, which operate based on the composition of score or noise predictions from the diffusion process, are limited to independent concepts (e.g., "a blonde girl" without "glasses") and fail to consider the interconnected nature of concepts in reality (e.g., "Mickey mouse eats ice cream" without "Disney characters"). Keeping the limitations in mind, we propose a novel framework, called PROTORE, to improve the flexibility of concept negation via test-time negative concept identification along with purification in the feature space. PROTORE works by incorporating CLIP's language-contrastive knowledge to identify the prototype of negative concepts, extract the negative features from outputs using the prototype as a prompt, and further refine the attention maps by retrieving negative features. Our evaluation on multiple benchmarks shows that PROTORE outperforms state-of-the-art methods under various settings, in terms of the effectiveness of purification and the fidelity of generative images.

## 1   Introduction

The family of diffusion models [1, 2] has achieved remarkable performance in image synthesis [3, 4, 5]. Recent advancements in text-conditional diffusion models [6, 7, 8, 9, 10] have further improved the ability to generate images with precise control over their content. In text-conditional diffusion models, text prompts[2] are used as input during the diffusion process to guide the creation of images that align with the desired content. Glide [6] and Semantic Diffusion Guidance (SDG) [7] have explored the use of pre-trained vision-language models like CLIP [11] to encode text conditions into latent features. Latent diffusion [8], including its large-scale implementation (Stable Diffusion), efficiently leverages and expands the design of latent vectors throughout the denoising process, where convolutional neural networks and cross-attention mechanisms merge multi-modal latent features.

---

*Corresponding authors: Song Guo, Junxiao Wang.

[2]Text prompts can take various forms, such as sentences, phrases, or single words, and serve as descriptions of desired image content. They provide a natural language interface to specify image attributes like style, color, and texture, enabling the generation of objects, scenes, and abstract concepts.

37th Conference on Neural Information Processing Systems (NeurIPS 2023).

Text-conditional diffusion models such as DALLE·2 [9] and Imagen [10] have unlocked the potential of generative models for various business applications with exceptional visual fidelity.

Nonetheless, the use of large-scale, web-scraped datasets like LAION [10, 12, 13] has raised ethical concerns among researchers. These unedited datasets often contain inappropriate and unauthorized content [12], posing risks. Users can manipulate text prompts to generate violent, pornographic, or copyright-infringing images, which can damage the reputation of the business model provider and lead to serious societal issues [14, 15, 16]. To address concerns about negative concepts generated by diffusion models, there is a growing interest in developing selective refusal techniques to confine the model's output space. Recently, researchers have been working on four approaches to reduce the generation of harmful content: filtering the dataset [6, 17], adversarial perturbations [16, 18, 19, 20], machine unlearning [21, 22], and implementing refusals during inference [23, 24]. Dataset filtering and perturbation-based methods primarily focus on preventive measures, making them less suitable for pre-trained well-established models. Unlearning-based methods often require modifications to global model parameters, which can limit scalability and hinder plug-and-play deployment capabilities. In contrast, refusals during inference time involve modifying the output of pre-trained models, making them more efficient for testing and deployment scenarios. Concept negation [22, 25] plays a crucial role in implementing such refusals by allowing the model to define and control its output space using language-based and human-understandable concepts. The current methods for concept negation perform on the composition of score or noise predictions from the diffusion process. However, these methods have limitations as they are confined to independent concepts and do not account for the interdependency of concepts in real-world scenarios.

**Our Contributions**. To address the aforementioned challenges, we propose a novel framework called PROTORE (Prototypical Refinement). Our approach enhances the flexibility of concept negation by introducing test-time negative concept identification and feature space purification. The PROTORE framework leverages CLIP's language-contrastive knowledge and follows a "Prototype, Retrieve, and Refine" pipeline. Here is a breakdown of the three steps involved: 1) Prototype: We utilize CLIP to encode a collection of text prompts obtained from social media platforms that express similar negative concepts. These encoded features are then aggregated into a comprehensive prototype feature, capturing the semantics of the negative concepts. 2) Retrieve: The negative prototype feature serves as a prompt to retrieve the model's output features that are correlated with the negative concepts. 3) Refine: We employ the retrieved negative features to refine the discriminative attention maps, purifying the influence of negative concepts in the feature space. By integrating these steps, our PROTORE framework offers a novel approach to concept negation, improving the flexibility and effectiveness of mitigating negative concepts in generative diffusion models. Moreover, this approach promotes scalability and enables easy deployment. Through comprehensive evaluations on multiple benchmarks, we demonstrate that PROTORE surpasses existing methods in terms of purification effectiveness and the fidelity of generated images across various settings.

## 2  Related Work

Recently, researchers have focused on addressing the generation of harmful content by exploring four approaches: dataset filtering [6, 17], adversarial perturbations [16, 18, 19, 20], machine unlearning [21, 22], and refusals during inference [23, 24].

**Dataset Filtering**. One straightforward approach to mitigate undesired image outputs in generative models is to filter out specific images from the training dataset. This can involve excluding certain image categories [17], such as those featuring people [6], or meticulously curating the data. However, dataset filtering has the drawback of being a costly solution to address issues discovered post-training, as retraining large models demands substantial resources.

**Adversarial Perturbations**. Another promising approach to protect images from being generated by large-scale generative models involves adding adversarial perturbations to raw images before uploading them on the internet or utilizing them in text-to-image AI systems like DALLE·2 and Stable Diffusion. While deep learning architectures are vulnerable to adversarial perturbations, previous research has primarily focused on their application in classification tasks [26, 27, 28]. Kos *et al.*[18] was the first to generate adversarial perturbations against deep generative models such as the variational autoencoder (VAE) and VAE-GAN. Liang *et al.*'s AdvDM [16] introduces adversarial perturbations into raw images by estimating the "attack vector" through Monte Carlo estimation.

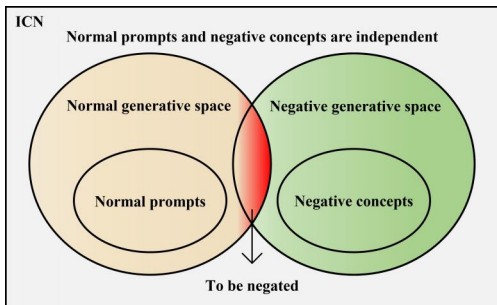
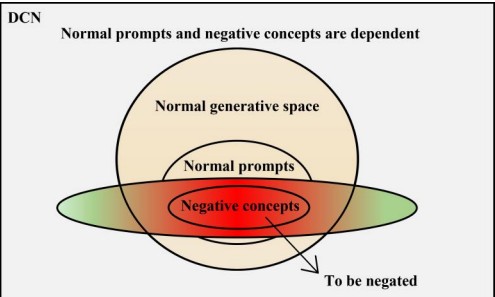

Figure 1: **A diagram illustrating the logical relationship between negative concepts and benign concepts in ICN (left) and DCN (right).** The generative space, representing the regions where diffusion models can generate valid samples, is visualized: the beige area represents valid samples conditioned on benign concepts, the green area represents valid samples conditioned on negative concepts, and the red area represents the overlap between these two spaces. In contrast to prior research [25, 31], we aim to remove the red area from the beige region.

Glaze [19] and PhotoGuard [20] apply perturbations that cause the model to confuse the perturbed image with an unrelated image or an image with a different artistic style. Similar to dataset filtering approaches, these methods typically adopt preventive measures to mitigate risks.

**Machine Unlearning**. The approach of machine unlearning [29, 30] involves the removal of specific knowledge from pre-trained models, including diffusion models. In their work, Moon *et al.*[21] tackled the challenge of unlearning a particular feature, such as a hairstyle, from pre-trained generative models. They devised an implicit feedback mechanism to identify the latent representation associated with the target feature and facilitate unlearning in the generative model. Gandikota *et al.*[22] proposed a method for fine-tuning diffusion model weights to eliminate specific concepts while minimizing interference with other concepts. In contrast to these approaches that globally modify model weights to unlearn undesirable outputs, our approach focuses on purifying negative outputs during inference, offering scalability and plug-and-play deployment capabilities.

**Refusals during Inference Time**. Previous studies have investigated methods of implementing refusals at inference time, as these methods are efficient for testing and deployment purposes. The core idea behind these approaches involves modifying the output after training using classifiers [23] or incorporating guidance into the inference process [24]. In contrast, our approach to refusal at inference time distinguishes itself by focusing on "concept negation" with a grounding in natural language. This enables defenders to easily specify normal and negative concepts in a text-conditional space, thereby enhancing the method's availability.

## 3 Concept Negation (NOT)

**Preliminaries**. A recent related work to our approach is the Composable Diffusion Models (CDM) [25], which provides an understanding of the diffusion model from the Energy-Based Model (EBM)'s perspective [31] and demonstrates how the additivity property of the EBM can be applied to diffusion generation. In this context, the generation process and scoring function of the diffusion model are referred to as $p_\theta^i(x_{t-1}|x_t)$ and $\epsilon_\theta^i(x,t)$, respectively. If we consider a single score function in a diffusion model as the learned gradient of the energy function in an EBM, the combination of diffusion models results in a score function denoted as $\sum_i \epsilon_\theta^i(x,t)$. Consequently, the generative process for combining multiple diffusion models can be expressed as follows:

$$p_{\text{CDM}}(x_{t-1}|x_t) = \mathcal{N}(x_t + \sum_i \epsilon_\theta^i(x_t,t), \sigma_t^2). \tag{1}$$

CDM aims to generate images based on a given set of concepts $\{c_1, c_2, \ldots, c_n\}$. To achieve this, each concept $c_i$ is represented as an individual diffusion model, and their score or noise predictions from the diffusion process are combined to generate the desired image. Drawing inspiration from EBMs, CDM introduces two combinatorial operators, namely conjunction (AND) and negation (NOT), to facilitate the combination of diffusion models. In the context of concept negation (NOT), consider the example of a user prompt "a blonde girl without glasses". In this case, the condition "a

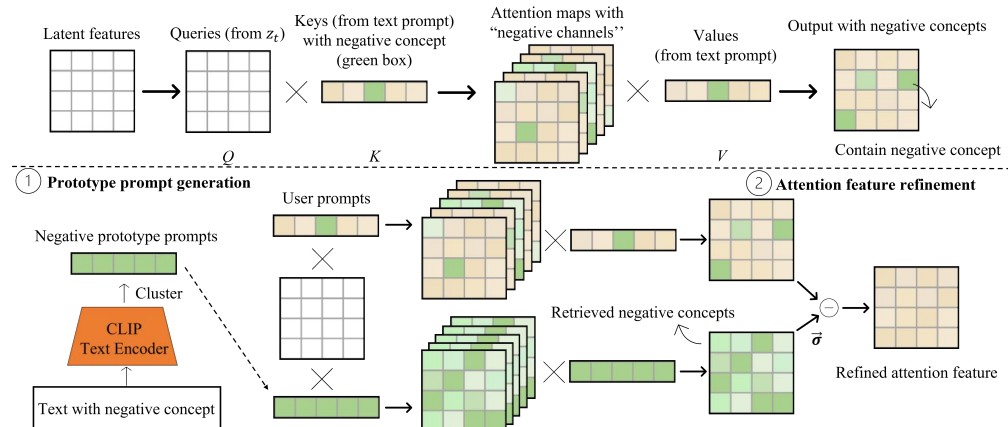

Figure 2: Method Overview. Top: The diffusion process entangles negative concepts, where cross-attentions align visual and textual embeddings, resulting in the concealment of negative concepts within attention maps (green box). Bottom: Our approach comprises two main steps: ① Generating negative prototype prompts using the CLIP text encoder. ② Refining the discriminative attention features by incorporating retrieved negative features, effectively purifying the influence of negative concepts in the feature space.

blonde girl" represents the benign concept $c_i$ that should be present in the generated images, while the text "glasses" represents the negative concept $c_j$ whose score or noise predictions need to be subtracted from the diffusion process. CDM's approach can combine pre-trained diffusion models within inference time without any additional training.

**Motivation**. Although CDM shows promise, it makes a significant assumption that the negative concept $c_j$ is independent of the benign concept $c_i$, which limits its flexibility in concept negation. For instance, consider a scenario where an image of "a Mickey Mouse eating ice cream" needs to be generated with "Mickey Mouse" or the broader condition "Disney character" as the negative concept. CDM would face challenges in handling such a situation. In our study, we propose a new taxonomy of concept negation, classifying CDM-like approaches as "independent concept negation" (ICN), while our work falls under the category of "dependent concept negation" (DCN). Figure 1 provides an intuitive illustration of the relationship between the ICN and DCN.

**Problem Definition**. Given a user prompt $c$ and a certain negative concept $\tilde{c}$, we aim to generate high-quality images $x$ describing $c$ with the absence of $\tilde{c}$. If we view concept negation in diffusion models as a probabilistic instantiation of logical operators applied to concepts. Formally, ICN factorizes the conditional generation as the following composed probability distribution:

$$\text{ICN: } p(\boldsymbol{x}|\boldsymbol{c}, \text{not } \tilde{\boldsymbol{c}}) \propto p(\boldsymbol{x}, \boldsymbol{c}, \text{not } \tilde{\boldsymbol{c}}) \propto p(\boldsymbol{x})\frac{p(\boldsymbol{c}|\boldsymbol{x})}{p(\tilde{\boldsymbol{c}}|\boldsymbol{x})}. \tag{2}$$

However, the formula (2) holds only when $c$ and $\tilde{c}$ are independent. Considering a realistic scenario, DCN formulates a more general conditional generation:

$$\text{DCN: } p(\boldsymbol{x}|\boldsymbol{c}, \text{not } \tilde{\boldsymbol{c}}) \propto p(\boldsymbol{x}, \boldsymbol{c}, \text{not } \tilde{\boldsymbol{c}}) \propto p(\boldsymbol{x})\frac{p(\boldsymbol{c}|\boldsymbol{x})}{p(\tilde{\boldsymbol{c}}|\boldsymbol{c}, \boldsymbol{x})}. \tag{3}$$

The natural ability of EBM [31, 32] and diffusion counterparts [25] to perform set-like composition through arithmetic on score or noise predictions would no longer hold when it comes to DCN. This implies that "A and not B" cannot be simply treated as the difference between log probability densities for A and B.

# 4 Method

Let $\mathcal{I}$ be an image generated using a diffusion model $\psi(\cdot)$ based on a user prompt $c$. The prompt may include up to $K$ pre-defined negative concepts $\tilde{c}_k, k = 1, 2, \cdots, K$, and our objective is to intervene in the image generation process to remove these concepts from $\mathcal{I}$. For example, if the prompt contains

"Mickey Mouse is eating ice cream", we may need to exclude the copyrighted content of "Mickey Mouse" to avoid legal issues while preserving other elements like "ice cream". This is crucial to prevent potential legal complications arising from using copyrighted or sensitive content. However, unlike previous approaches such as ICN [31, 25] and inpainting [8], we cannot rely on user-defined prompts or masks to determine where to remove or reconstruct. One intuitive approach is to create a list of prohibited words and remove them from the user's prompt whenever they appear. However, this approach has limitations, as synonyms may convey the same concept, and the list may not cover all possible negative expressions. Additionally, it may impede the normal use of the model. To overcome these challenges, we propose a plug-and-play concept negation method that enables text-conditional refusals for diffusion models during inference. The proposed method is illustrated in Figure 2.

**Prototype Prompt Generation**. To encode text prompts, we utilize the CLIP (Contrastive Language-Image Pre-training) text encoder $\varphi(\cdot)$ [11], which is a state-of-the-art model of vision-language representation learning. CLIP is pre-trained on a vast corpus of text and images using a contrastive loss function, which encourages the model to map semantically similar text and images to nearby points in a shared embedding space. However, explicitly enumerating all possible prompt words of negative concepts $\tilde{c}_k$ can be challenging. Instead, we crawl a set of text prompts $\tilde{C}_k$ expressing similar negative concepts from social media platforms, aiming to elicit the $k$-th negative concept in the generated images. We then rely on the zero-shot classification's capability of CLIP to encode the negative text prompts into multiple high-dimensional features. These prompts are then aggregated (clustered) into a single prototype prompt $\tilde{c}_k^*$ representing the respective negative class:

$$\tilde{c}_k^* = \texttt{cluster}(\varphi(\tilde{C}_k)). \tag{4}$$

We employ three distinct clustering methods to derive the prototype prompts. In the case of single-label to single-class refusals (e.g., ImageNet), we utilize the embedding of the corresponding label as the prototype prompt. For multiple-labels to multiple-class refusals (e.g., I2P datasets), the clustering center is computed using K-means, which then serves as the prototype prompt. As for multiple dependent concept refusals (as depicted in Figure 4), we combine all the concepts using commas or the word "and" to form the prototype prompt.

Prior studies on image editing [33] have shown that diffusion models utilize cross-attention layers to combine visual and textual features, resulting in the generation of spatial attention maps for each textual token. As shown in the top of Figure 2, the visual features of a noisy image $z_t$ are projected to a query matrix via a linear layer $\ell_Q(\cdot)$, and $Q = \ell_Q(z_t)$. Similarly, the textual feature $\varphi(c)$ is projected to a key matrix $K = \ell_K(\varphi(c))$ and a value matrix $V = \ell_V(\varphi(c))$ through linear layers $\ell_K(\cdot), \ell_V(\cdot)$. The attention maps $M$ are then calculated as follows:

$$M = \text{Softmax}\left(\frac{QK^T}{\sqrt{d}}\right) = [\text{attn}_1, \text{attn}_2, \cdots, \text{attn}_l, \cdots, \text{attn}_L] \in \mathbb{R}^{C \times (H \times W) \times L}, \tag{5}$$

where $d = H \times W$ represents the dimension of the key and value projection layers. $C$ and $L$ denote the number of attention heads and tokens in a single sequence, respectively. Channel $\text{attn}_l \in \mathbb{R}^{C \times H \times W}$ can be squared and visualized as the $l$-th token in the text prompt $c$. As a result, negative concepts in $\text{attn}_l$ may appear in the weighted output features $A = M \cdot V$. We have noticed that negative concepts are frequently intertwined with normal concepts in the attention maps, which may explain why EBM and CDM are not as efficient. Furthermore, we lack prior knowledge about the user prompt, including which negative class it belongs to and where the negative token is located. This makes it difficult to use existing image editing methods that depend on explicit guidance signals for concept negation.

**Attention Feature Refinement**. We employ the prototype prompt generated using Equation (4) to retrieve the negative concepts present in the output features. Subsequently, using Equation (5), we calculate the negative attention maps for the prototype feature $\tilde{c}_k^*$:

$$\widehat{M}^k = \text{Softmax}\left(\frac{Q\widehat{K}^T}{\sqrt{d}}\right) = \text{Softmax}\left(\frac{\ell_Q(z_t)\ell_K(\tilde{c}_k^*)^T}{\sqrt{d}}\right). \tag{6}$$

Intuitively, $\widehat{M}^k$ highlights the $k$-th negative concepts present in the noisy image $z_t$. We proceed to obtain the negative features, denoted as $\widehat{A}^k$, through $\widehat{A}^k = \widehat{M}^k \cdot \widehat{V}^k$, where $\widehat{V}^k = \ell_V(\tilde{c}_k^*)$. Finally, to eliminate the identified negative features, we compute the refined attention features $A^*$ by:

$$A^* = A - \sum_k \sigma_k \cdot \widehat{A}^k, \tag{7}$$

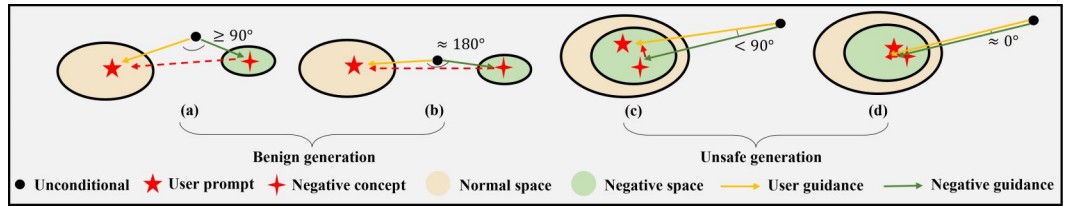

Figure 3: Understanding the mechanism behind PROTORE's success. By using user prompts (represented by the yellow arrow) and prototype negative prompts (represented by the green arrow), we guide the diffusion process of Gaussian noise toward the intended space. We analyze the effectiveness of attention feature refinement in two common generation scenarios: benign generation and unsafe generation. These scenarios are determined by the relationship between user prompts and negative concepts. Our proposed attention feature refinement technique (indicated by the red dashed arrow) successfully redirects the diffusion process away from the negative space, ensuring convergence towards the normal space in both scenarios.

where coefficient $\sigma_k$ controls the refinement step size.

The resulting attention feature refinement progress is intuitively visualized in Figure 3. In benign image generation scenarios (a) and (b), user prompts remain dissociated from negative concepts. We can observe that the refined diffusion process (indicated by the red dashed arrow) gradually converges to normal space, thereby affirming that attention feature refinement does not interfere with the generation of user-specified images. In contrast, the user prompts associated with unsafe image-generation scenarios involve negative concepts. In such cases, refinement of the diffusion process leads to a gradual shift away from the negative space, facilitating the effective negation of undesirable concepts from the generated images. In order to facilitate comprehension, two specific cases are examined in (b) and (d) where user prompts and negative concepts are either opposite or identical, respectively. Sub-figure (b) depicts a scenario in which the angle between the user and negative guidance is approximately 180 degrees. Under this circumstance, attention feature refinement modifies the diffusion process by increasing the step size along the user prompt direction, with no disturbance to image generation. In contrast, in sub-figure (d), the angle between the user and negative guidance approaches zero degrees. In this case, attention feature refinement directs the diffusion process away from negative guidance along the user guidance or in the opposite direction. Our intuition is verified by the experimental results and the discussion of method limitations.

Our proposed algorithm is formally presented in the Appendix, which consists primarily of two steps. Firstly, negative prototype prompts are generated using Equation (4), which can be completed offline. At inference time, diffusion models denoise a given corrupted input sample (Gaussian noise $z_T$) iteratively by estimating the conditional probability distribution that approximates the target distribution of the clean sample $z_0$. At each timestamp $t$, the attention feature $A_t$ is calculated based on the user prompt $c$, which may contain up to $K$ classes of negative concepts. We then sequentially retrieve and remove negative concepts using Equation (7).

## 5 Experiments

In this section, we empirically evaluate the effectiveness of our proposed PROTORE. The refinement step size $\sigma$ is set to 1.0 in our experiments unless specified otherwise. We benchmark our approach against the following baseline method: Stable Diffusion v2.1 (SD) [8]; composable diffusion models (CDM) [25], to adapt this method to our experiment, we configure the negative concepts as the unconditional conditioning prompt; safe latent diffusion (SLD) [24], both CDM and SLD study refusals at inference time; erased stable diffusion (ESD) [22], an approach for eliminating specific concepts by fine-tuning; Stable Diffusion with negative prompts (SD-Neg), an intuitive method that manually adds negative prompts, such as "without bear", behind the user prompt.

### 5.1 Single-Concept Refusals

**ImageNet subset**. We first investigate the performance of single-concept refusal through numerical results. Specifically, we choose one class from ImageNet as the negation target. To measure the

Table 1: Quantitative refusal results on Imagenette subset.

| Class Name | Accuracy of erased class ↓ | | | | | | Accuracy of other classes ↑ | | |
|---|---|---|---|---|---|---|---|---|---|
| | SD | SD-Neg | SLD [24] | CDM[25] | ESD [22] | PROTORE | SD | ESD | PROTORE |
| cassette player | 13.0 | 11.6 | 0.4 | 3.4 | 0.60 | **0** | 91.71 | 64.5 | **84.76** |
| chain saw | 91.0 | 88.0 | 5.2 | 4.6 | 6.0 | **0.2** | 83.04 | 68.2 | **74.84** |
| church | 98.0 | 98.0 | 62.8 | 83.0 | **54.2** | 85.2 | 82.27 | 71.6 | **80.16** |
| gas pump | 96.0 | 96.2 | 31.6 | 18.6 | 8.6 | **0** | 82.49 | 66.5 | **67.78** |
| tench | 89.8 | 91.8 | 66.4 | 39.6 | 9.6 | **1.6** | 83.18 | **66.6** | 56.58 |
| garbage truck | 79.6 | 79.6 | 34.8 | 21.4 | 10.4 | **0** | 84.31 | 51.5 | **70.58** |
| English springer | 99.6 | 99.2 | 95.4 | 12.8 | 6.2 | **0** | 82.09 | 62.6 | **74.71** |
| golf ball | 90.2 | 86.8 | 81.8 | 17.6 | 5.8 | **0.6** | 83.13 | 65.6 | **78.42** |
| parachute | 83.8 | 80.2 | 58.4 | 26.2 | 23.8 | **7.8** | 83.84 | 65.4 | **81.80** |
| French horn | 97.4 | 98.4 | 92.8 | 28.6 | 0.4 | **0** | 82.33 | 49.4 | **73.22** |
| Average | 83.84 | 82.89 | 52.96 | 25.58 | 12.6 | **9.54** | 83.84 | 63.2 | **74.28** |

effectiveness of erasing the targeted class, we generate 500 images with the prompt "an image of a [*class name*]". Then, our assessment entails examining the top-1 prediction accuracy of a pre-trained Resnet-50 Imagenet classifier. Following the same setting in ESD [22], we select the Imagenette subset that consists of ten readily recognizable classes.

The left side of Table 1 presents quantitative results comparing the classification accuracy of the erased class using the original Stable Diffusion model and four refusal methods. The proposed method shows higher performance in most classes, which highlights the effectiveness of attention corrections within the Stable Diffusion. However, existing methods have certain drawbacks: SD-Neg is only able to marginally remove the specified class, indicating that it is challenging for the Stable Diffusion to understand the explicit "without" command in the prompt. SLD introduces auxiliary guidance to adjust the noise prediction of U-Net in SD, enabling its suitability for the localized image detail retouching task. However, the network's effectiveness in object removal appears limited. The inefficiency of CDM can be attributed to the incapability of the solution to the ICN problem to generalize well to the DCN problem. Despite exhibiting moderate effectiveness, ESD incurs additional training resources for fine-tuning the Stable Diffusion, requiring training multiple models for each class to obtain distinct copies. For instance, ESD would mandate ten fine-tuned Stable Diffusion models, each responsible for erasing a single class in Imagenette. Consequently, the storage capacity necessary to accommodate the gradual increase in negative classes would escalate. Instead, our proposed plug-and-play approach offers a training-free solution with the ability to flexibly switch between different target classes.

The refusal methods employed for erasing target class concepts should not impede the generation of images for other classes. The average accuracy of producing images for the remaining nine non-target classes after removing the target class is illustrated on the right side of Table 1. Our proposed method preserves the model's capacity for generating benign images better compared to ESD. These results suggest that fine-tuning-based machine unlearning for the target class comes at the cost of sacrificing the original model's capacity for image generation.

**Inappropriate Image Prompts (I2P) benchmark dataset** [24] contains 4703 toxic prompts assigned to at least one of the following categories: hate, harassment, violence, self-harm, sexual, shocking, illegal activity. We generate five images for each prompt and employ the Q16 [34] and NudeNet [3] classifiers to quantify the proportion of generated inappropriate images. The toxic description provided by I2P serves as the prototype negative prompt for comparative analysis. Table 2 displays the quantitative refusal results of three methods. Our proposed PROTORE can considerably diminish the possibility of generating inappropriate images, demonstrating the effectiveness in confining the output space of negative concepts in the model.

---

[3]https://github.com/notAI-tech/NudeNet

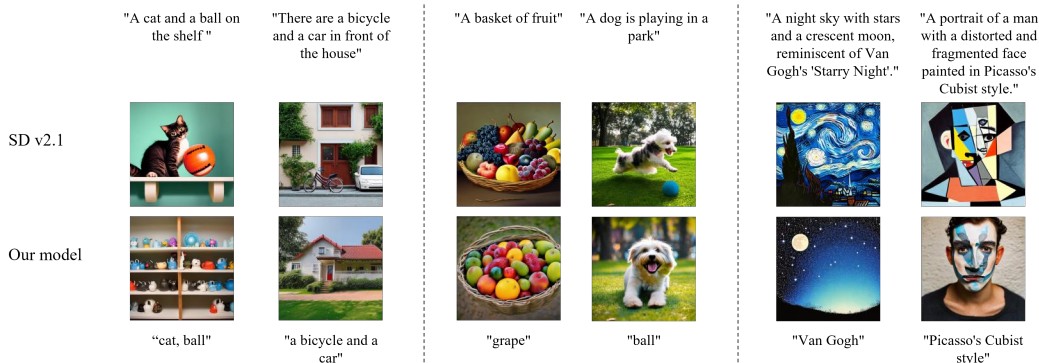

Figure 4: Qualitative refusal results on multiple concepts (left), implicit concepts (middle), and artistic styles (right). Further examples can be found in Appendix-D.

Table 2: Quantitative refusal results on I2P benchmark [24].

| Class Name | Inappropriate Probability ↓ | | | |
|---|---|---|---|---|
| | SD | ESD | SLD | PROTORE |
| Hate | 0.40 | 0.17 | 0.20 | **0.10** |
| Harassment | 0.34 | 0.16 | 0.17 | **0.07** |
| Violence | 0.43 | 0.24 | 0.23 | **0.09** |
| Self-harm | 0.40 | 0.22 | 0.16 | **0.09** |
| Sexual | 0.35 | 0.17 | 0.14 | **0.08** |
| Shocking | 0.52 | 0.16 | 0.30 | **0.10** |
| Illegal activity | 0.34 | 0.22 | 0.14 | **0.11** |
| Average | 0.39 | 0.19 | 0.19 | **0.09** |

Table 3: Image Fidelity Performance on COCO 30k dataset.

| Method | FID-30k |
|---|---|
| SD | 14.50 |
| SLD | 16.90 |
| ESD | 13.68 |
| PROTORE | 16.80 |

## 5.2 Complex Concept Refusals

We then evaluate the refusal capability of PROTORE in three complex scenarios, namely multi-concept refusals, implicit concept refusals, and artistic style refusals to demonstrate its effectiveness. Qualitative results are shown in Figure 4. Figure 4 shows qualitative results. Our approach effectively removes multiple concepts (e.g., "cat" and "ball", "bicycle" and "car")[4] simultaneously and offers deployers the flexibility to adjust removed concepts (add or delete) according to policies and regulations, as demonstrated on the left side of Figure 4. The middle of Figure 4 highlights the effectiveness of PROTORE in removing implicit concepts that may accidentally appear in generated images despite not being explicitly mentioned in the prompt [24], such as removing the "grape" in "a basket of fruit" or the "ball" in "a dog is playing in a park". The right side of Figure 4 demonstrates the ability of our approach to remove specific artistic styles, ensuring content creators using the diffusion model do not infringe on copyright laws. The proposed PROTORE approach showcases promising results in all three scenarios, highlighting its efficacy in complex, real-world applications.

## 5.3 Refusal Steps Setting

In this study, we examine the impact of the choice of diffusion steps, referred to as "refusal steps", on PROTORE. The initial diffusion steps have a greater influence on the generated image's structure, whereas the later steps have a lesser effect on the content. As illustrated in Figure 5, our findings demonstrate that using our method beyond the initial steps (e.g., starting from the 10th step) preserves the underlying image structure. Conversely, implementing our method during the first steps results in significant alterations to the image's appearance and the removal of specific content. More fine-grained results can be found in the Appendix.

---

[4] In our experiments, we did not use inappropriate concepts such as infringement or nudity as prompts; instead, we used some common concepts to demonstrate the refusal effectiveness.

User prompt: "a professional photograph of an astronaut riding a triceratops" , negative concept "a triceratops"

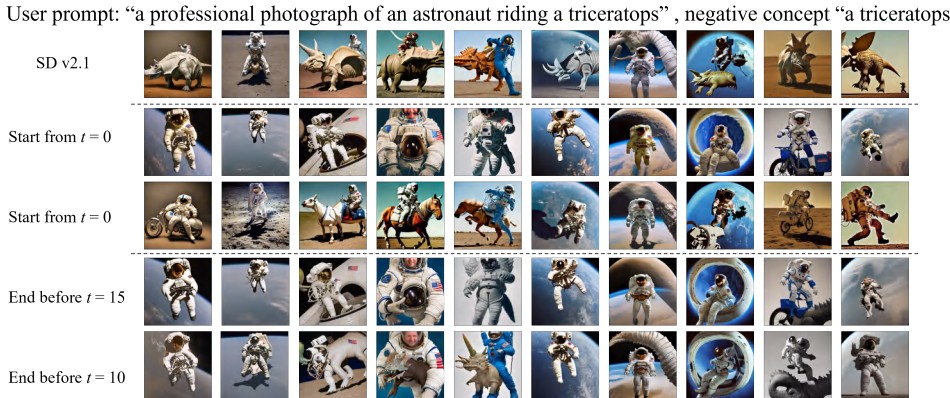

Figure 5: Varying the diffusion steps at which PROTORE is applied will impact the generated output. Specifically, "Start from $t$" indicates applying PROTORE during diffusion steps $t$ to 50, while "End before $t$" refers to applying PROTORE during diffusion steps 0 to $t$.

User prompt = "An image of a church"     Negative concept = "church"

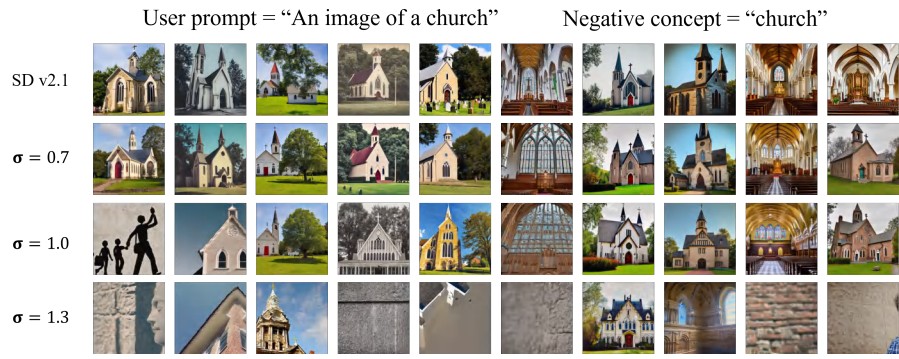

Figure 6: Cases of incomplete concept negation with our method. The refusal effect gradually amplifies with the increasing of the refinement step size $\sigma$.

## 5.4 Image Fidelity Preserving

We investigate the impact of the proposed concept negation techniques on image fidelity to ensure that the erased model maintains its ability to generate safe content effectively. It is desirable for the methods employed to have no adverse effects on appropriate images. To this end, we follow prior work [24, 22] on generative text-to-image models and evaluate the COCO FID-30k scores of SD and the three additional methods, as presented in Table 3. Fréchet Inception Distance (FID) is widely utilized to assess the quality of the generated samples. This is accomplished by utilizing an inception network to extract relevant features from both real images and generated samples. The FID metric evaluates the similarity between the two distributions by measuring their distance. We employed inference guidance of 7.5 in our experiments. Our proposed approach demonstrates superior image fidelity performance compared to SLD and Stable Diffusion when applied to COCO 30k images. The experimental results suggest that our proposed method can effectively enable test-time refusals without compromising the normal generative capacity of the model. This indicates that PROTORE serves as a seamless plug-and-play operator that can be smoothly incorporated into text-conditional diffusion models.

## 5.5 Limitations

Our findings reveal that our proposed method performs better in eliminating small objects compared to larger objects. When it comes to removing a target concept that covers almost the entire image, such as "church", our method falls short of completely eliminating it. This observation aligns with the reasoning depicted in Figure 3-(d), where we ascribe this outcome to the small refinement step size, which insufficiently directs the generated image from negative space to normal space in a 50-step

diffusion process. Supporting this claim, Figure 6 illustrates that the refusal effect gradually amplifies with the increasing of the refinement step size $\sigma$.

Another limitation is that discrepancies or variations in semantic descriptions within the same class can significantly impact the computation of cluster centers. For instance, when considering the topic of "violence," there exist numerous descriptions that encompass different facets, such as the intentional use of physical force or power, potential harm, and psychological consequences, among others. The presence of such diverse descriptions poses challenges in deriving an appropriate prototype prompt using the K-means algorithm.

In the context of this paper, our focus lies on investigating relatively uncomplicated scenarios that involve the rejection of well-defined objects or styles. Nonetheless, we are aware of the necessity to address more intricate situations, where concepts with complex abstract semantics are involved. As part of our future research efforts, we will explore methodologies to effectively eliminate or handle these intricate concepts, thereby enhancing the robustness and accuracy of our approach.

It is imperative to acknowledge that our current approach may exhibit diminished effectiveness when encountering user prompts that involve intricate negation with compositional or relational information. The generated images may, therefore, exhibit instances of attribution leaks, wherein characteristics of one entity, such as a horse, mistakenly manifest in another, like a person, as well as occurrences of missing objects, erroneously omitting human or equine elements, and so forth. We recognize that the capabilities of CLIP (Contrastive Language-Image Pretraining) in processing textual prompts do have an impact on the overall performance of the methods presented in our paper. We will discuss these limitations in our paper and foster further investigation in future work.

# 6   Conclusion

This paper proposes a novel approach, PROTORE (Prototypical Refinement), to address the challenges of unsafe generation in diffusion models. Unlike previous methods that rely on the composition of score or noise predictions from the diffusion process and fail to effectively remove negative concepts during inference, PROTORE introduces test-time negative concept identification and feature space purification to enhance the flexibility of concept negation. PROTORE works by incorporating CLIP's language-contrastive knowledge to identify the prototype of negative concepts, extract the negative features from outputs using the prototype as a prompt, and further refine the attention maps by retrieving negative features. As a critical consideration, we suggest eliminating the negative features in the cross-attention layers for text-conditional refusals which merge visual features and textual guidance. Our PROTORE framework effectively mitigates negative concepts across various settings and demonstrates scalability and ease of deployment. Comprehensive evaluations on multiple benchmarks demonstrate the superiority of PROTORE over existing methods in achieving better purification effectiveness and preserving image fidelity.

## Acknowledgments and Disclosure of Funding

This research was supported by fundings from the Key-Area Research and Development Program of Guangdong Province (No. 2021B0101400003), Hong Kong RGC Research Impact Fund (No. R5060-19, No. R5034-18), Areas of Excellence Scheme (AoE/E-601/22-R), General Research Fund (No. 152203/20E, 152244/21E, 152169/22E, 152228/23E), Shenzhen Science and Technology Innovation Commission (JCYJ20200109142008673).

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
