# OpenReview forum: "Towards Test-Time Refusals via Concept Negation"
_NeurIPS.cc/2023/Conference — NeurIPS 2023 poster_

### Official Review · Reviewer_WXn9 · 2023-06-27

**Soundness:** 4 excellent
**Presentation:** 2 fair
**Contribution:** 3 good
**Rating:** 7
**Confidence:** 2

**Summary:**

The paper proposes a method to remove a negative concept in a text-to-image diffusion model during inference time; the approach is different from prior work, which assumed that the benign concepts and the removed concepts are independent and added a penalty score; instead, this paper proposes to remove the concepts by 1) extract features from prompts that are known to be negative, and 2) use the extracted features to refine the attention maps. Results on object removal benchmarks look promising.

**Strengths:**

The results seem strong, i.e., the method is outperforming previous methods according to Table 1. However, I'm not an expert in this field so I am not confident whether the evaluation is reliable.

**Weaknesses:**

- The paper reads confusing to me (probably because I did not work on text-to-image diffusion methods). This might not be a weakness depending on the audience, but its contribution might not deliver to the broader audience as it is currently written.
- I am unsure about the motivation of the method: if we want to remove certain concepts, can we either 1) prompt ChatGPT to rewrite the prompt to remove the concept, or 2) change the prompt to explicitly mention that [original prompt] + "please do not include XXX"? This seems to me the most natural/straightforward approach, so perhaps it is useful to include this as a baseline.
- I am slightly confused about the desired behavior. What should "Mickey Mouse eating ice-cream" - "Disney Character" look like? Would it be just the ice-cream

**Questions:**

078: unclear why “adding adversarial perturbations” would be able to protect images from being generated (speaking as a reader not familiar with adversarial training). Would be helpful to add a sentence to talk about the underlying intuition. Not a critical concern.

Equation 2: This might be a stupid question, but why does $p(x, c, not \tilde(c)) \propto p(x)p(c|x) / p(\tilde(c)| x)$? Does this follow from probabilistic derivation, or it’s the definition proposed by the paper? If it’s the former, can you derive it in the appendix why it is true (without using the \propto symbol but state the equation fully by including the normalization factors excluded here); if it’s the later, can you briefly justify in the paper why it is a useful definition?

151: rather than creating a list of prohibited words, can we simply ask ChatGPT to detect and remove similar concepts and still make it a coherent prompt?

190 & figure 3: sorry I’m confused about the figure, which seems important in conveying the core idea of the paper. For example, in a), if the red dashed line represents the diffusion process, does it mean that the image changes from an unsafe image to a safe image? Why would the model generate an unsafe image in the first place under a safe user prompt? If an image is considered safe, then it should be in the orange circle, but there is nothing in c) and d) in the orange circle.

I think the authors probably do have a good understanding of what they are doing but I cannot really tell what it is based on the figure.


**Limitations:**

The authors discussed the limitation of their work.

---

> ### Author Rebuttal · Authors · 2023-08-07
>
> Thank you for your valuable feedback! Below we address your questions and concerns. Please feel free to post additional comments if you have further questions.
> 1. Deep learning architectures can be susceptible to adversarial perturbations. The introduction of deliberate perturbations can disrupt the internal processes of the targeted diffusion model, leading the model to misinterpret the perturbed image as unrelated content. This vulnerability becomes particularly significant when adversaries attempt to exploit the diffusion model for editing images to generate illegal content. By introducing perturbations to such images, it becomes possible to prevent the images from being altered to achieve the adversary's intended result. For a concrete visualization example demonstrating these concepts, please refer to Figure 1 in the paper titled "Raising the Cost of Malicious AI-Powered Image Editing."
> https://arxiv.org/pdf/2302.06588.pdf.
> 2. Thank you for bringing this to our attention. The Equation (2) as you mentioned was initially introduced in the works of [1-2]. We will provide its derivation and explanations in the appendix. Diffusion models demonstrate the ability to compute both unconditional generation ($p(x)$) and conditional generation given a prompt ($p(x|c)$). Notably, when Equation (2) is satisfied, the intricate multi-conditional generation process, involving conditions such as $c$ and $\hat{c}$, can be decomposed into a combination of distinct single-conditional generations, simplifying the overall generation process.
>
> [1] Du, Yilun, Shuang Li, and Igor Mordatch. "Compositional visual generation with energy based models." Advances in Neural Information Processing Systems 33 (2020): 6637-6647.
>
> [2] Liu, Nan, et al. "Compositional visual generation with composable diffusion models." European Conference on Computer Vision. Cham: Springer Nature Switzerland, 2022.
>
> 3. The strategy of eliminating negative concepts through rephrasing user prompts using a large language model like ChatGPT holds promise as a viable approach. However, the key hurdle lies in devising suitable rephrasing prompts. It is essential to carefully design these prompts to achieve the desired outcome effectively.
> Nevertheless, we must acknowledge a notable limitation of this approach. While attempting to condition images on normal prompts, the presence of negative concepts may inadvertently occur (as demonstrated in Figure 4, middle). Regrettably, relying solely on ChatGPT is insufficient to address this problem effectively.
> We also evaluate the pattern [original prompt] + "please do not include XXX". A comprehensive results will be provided in the appendix, offering a more complete understanding of its potential efficacy.
>
> 4. Figure 3 illustrates two generation scenarios: benign generation (user prompts without negative concepts) and unsafe generation (user prompts containing negative concepts). The red dashed line represents the refinement direction of our approach. We use this direction to show that our method has no impact on benign generation while effectively removing negative concepts during unsafe generation. In the case of benign generation, the refinement direction points to the normal space. Even when there are no negative concepts to remove, our method does not interfere with the regular image generation process. On the other hand, for unsafe generation, the refinement direction guides the diffusion process away from the negative space and towards the normal space. This indicates that our approach can successfully eliminate negative concepts after several iterations.
>
> For weakness 3:
>
> Consider the text prompt "mickey mouse eating an ice cream" and a pre-defined negative concept "N = {mickey mouse}". The remaining part of the prompt becomes the positive concept "P = {eating an ice cream}". A good generative result in this case would include ice cream but not mickey mouse. As we do not primarily focus on personalized generation for specific contents, a non-mickey mouse outcome or even a missing object is acceptable. Our primary objective is to prevent the generation of negative content and ensure that the remaining positive concepts given by the prompt are generated accurately. However, discussing what content should replace the removed negative concepts is beyond the scope of this paper.

---

> > ### Comment · Reviewer_WXn9 · 2023-08-10
> > **The response clarifies all my confusions.**
> >
> > Thanks for your response. It clarifies my confusions.
> >
> > (I would love to increase the score to 7 though I do not know how to do that) --> nvm just increased the score.

---

### Official Review · Reviewer_e3Lc · 2023-07-02

**Soundness:** 3 good
**Presentation:** 2 fair
**Contribution:** 4 excellent
**Rating:** 7
**Confidence:** 3

**Summary:**

Generative models require refusal techniques to limit their output and uphold ethical and copyright standards. 'Concept negation' is a promising solution, yet current methods have limitations, such as only accommodating independent concepts without considering their interconnected nature. This paper proposes PROTORE, a novel framework that enhances concept negation by identifying and purifying negative concepts during testing. PROTORE leverages the language-contrastive knowledge of CLIP to refine the attention maps and extract negative features. Evaluations show PROTORE outperforms existing methods in efficiency of purification and image quality.

**Strengths:**

- The proposed PROTORE is plug-and-play method that can be easily adopted in practice. It is able to refine the negation concept from user prompt during test-time without additional training.
- The proposed method shows strong results comparing other methods on Imagenette and (I2P) benchmarks. It also decently maintain performance on other classes in controllable comparision in Table 1 indicating its superior to ESD.
- PROTORE is also able to maintain good FID comparing to other methods.

**Weaknesses:**

- The proposed method might be limited or upper bounded by the general language ability of CLIP. It have been known that CLIP hold limited composition or grammar understanding of text input. This means that the proposed method might not be able to correctly negate concept like "human riding a horse" when user prompt is "a human and a horse". It could be good to have those studies or discussion briefly in the paper such as in the limitation section.
- The paper does not release the crawled negation prompts datasets. The method cluster the prompt into k concepts. It's unknown how the clustering method or the crawling method affect the effectiveness of the PROTORE method.

**Questions:**

How effective do you think the current method could handle complex negation with composition or relation information?
How will crawling or clustering method affect PROTORE?

**Limitations:**

The paper explained and included the limitation section.

---

> ### Author Rebuttal · Authors · 2023-08-07
>
> Thank you for your valuable feedback! Below we address your questions and concerns. Please feel free to post additional comments if you have further questions.
>
> 1. Thank you for bringing this to our attention. It is imperative to acknowledge that our current approach may exhibit diminished effectiveness when encountering user prompts that involve intricate negation with compositional or relational information. The generated images may, therefore, exhibit instances of attribution leaks, wherein characteristics of one entity, such as a horse, mistakenly manifest in another, like a person, as well as occurrences of missing objects, erroneously omitting human or equine elements, and so forth. We recognize that the capabilities of CLIP (Contrastive Language-Image Pretraining) in processing textual prompts do have an impact on the overall performance of the methods presented in our paper. We will discuss these limitations in our paper and foster further investigation in the future work.
> 2. We employ three distinct clustering methods to derive the prototype prompts. In the case of single-label to single-class refusals (e.g., ImageNet), we utilize the embedding of the corresponding label as the prototype prompt. For multiple-labels to multiple-class refusals (e.g., I2P datasets), the clustering center is computed using K-means, which then serves as the prototype prompt. As for multiple dependent concepts refusals (as depicted in Figure 4), we combine all the concepts using commas or the word "and" to form the prototype prompt.
>
>     It is crucial to acknowledge that discrepancies or variations in semantic descriptions within the same class can significantly impact the computation of cluster centers. For instance, when considering the topic of "violence," there exist numerous descriptions that encompass different facets, such as the intentional use of physical force or power, potential harm, and psychological consequences, among others. The presence of such diverse descriptions poses challenges in deriving an appropriate prototype prompt using the K-means algorithm.
>
>     In the context of this paper, our focus lies on investigating relatively uncomplicated scenarios that involve the rejection of well-defined objects or styles. Nonetheless, we are aware of the necessity to address more intricate situations, where concepts with complex abstract semantics are involved. As part of our future research efforts, we will explore methodologies to effectively eliminate or handle these intricate concepts, thereby enhancing the robustness and accuracy of our approach.

---

> > ### Comment · Reviewer_e3Lc · 2023-08-18
> >
> > Thank you for the response. The response addressed my concerns.

---

### Official Review · Reviewer_p7ZQ · 2023-07-10

**Soundness:** 4 excellent
**Presentation:** 4 excellent
**Contribution:** 4 excellent
**Rating:** 7
**Confidence:** 3

**Summary:**

This paper tackles the problem of concept negation in the context of image editing (i.e., excluding user specified concepts in the generated image). To this end, this paper proposes ProtoRe, which consists of three steps: 1) first a collection of negative prompts are encoded using CLIP and eventually aggregated, 2) retrieve the model’s output features based on the negative concepts, and 3) refine the attention map using the retrieved negative features.

In the experiments, this approach is evaluated in terms of accuracy on erased classes (lower is better) and other classes (higher is better). Overall, this approach outperforms the baselines such as Stable Diffusion, composable diffusion model, and safe latent diffusion (with a few exceptions).


**Strengths:**

- This work tackles an important problem of controlling contents generated by text-to-image generation models. The approach is kept simple yet effective on the target task.
- The experimental results (both quantitative and qualitative results) suggested the effectiveness of this approach.


**Weaknesses:**

- Evaluating generated outputs can be challenging, so it would be beneficial to include more detailed discussions and analyses of both successful and unsuccessful cases.


**Questions:**

- This paper might be related: Imagen editor and editbench: Advancing and evaluating text-guided image inpainting (https://openaccess.thecvf.com/content/CVPR2023/papers/Wang_Imagen_Editor_and_EditBench_Advancing_and_Evaluating_Text-Guided_Image_Inpainting_CVPR_2023_paper.pdf)

---

> ### Author Rebuttal · Authors · 2023-08-07
>
> Thank you for your valuable feedback!
>
> The paper "Imagen Editor and EditBench: Advancing and Evaluating Text-Guided Image Inpainting" centers around a distinct task known as text-guided image editing (TGIE). Unlike complete image generation, TGIE involves the editing of pre-existing or captured visuals. In this process, the user provides three inputs: 1) the image to be edited, 2) a binary mask specifying the edit region, and 3) a text prompt—all of which guide the output samples. Due to this fundamental difference, we recognize that the relationship between TGIE and our approach is limited. Nevertheless, both TGIE and our method share the common objective of achieving efficient, automated, and controllable visual generation.
>
> Please feel free to post additional comments if you have further questions.

---

### Official Review · Reviewer_2mKf · 2023-07-21

**Soundness:** 2 fair
**Presentation:** 2 fair
**Contribution:** 3 good
**Rating:** 4
**Confidence:** 3

**Summary:**

Paper is concerned with stable diffusion of images guided by a prompt but with stipulated concepts not present. In particular it is concerned with situations where it is unclear how the prompt may be visualized without visualizing the banned concepts.

Method works by incrementally pushing the generated image latent feature away from a feature vector representing the negative concepts. Where the negative feature vector has been obtained by expanding the negative concept prompts using a relevant corpus.



**Strengths:**

A very important live topic urgently requiring effective solutions.

Attempts to deal with the realistic scenario where the there is considerable overlap between the positive and negative prompts.

Some results are impressive.

**Weaknesses:**

Paper is not easy to follow. In particular, figures 1 and 3 are not clarifying.

Insufficient clarifying discussion on what the system should be outputting when positive and negative prompts are overlapping.

Results insufficiently discussed - for example readers needs guidance to appreciate fig 5.


**Questions:**

On the lack of clear aim of the system: what would a good result would be for the exemplar 'mickey mouse eating an ice cream' - 'mickey mouse'. Would a good result  be a non-mickey mouse eating an ice cream, for example 'minnie mouse', or a non-cartoon mouse, or a person rather than a mouse. Visualizing P alone is straightforward (maximize semantic relatedness to P); avoiding N is similarly (minimize s.r. to N); but together is not clear and surely some balancing is needed and can this be safely done?

**Limitations:**

Any system such as this has the potential for mis-use. Social manipulation could be achieved by covert banning of certain concepts e.g. 'police violence'. The fact of mis-use and the need to balance against potential benefits should be noted.

---

> ### Author Rebuttal · Authors · 2023-08-07
>
> Thank you for your valuable feedback! Below we address your questions and concerns. Please feel free to post additional comments if you have further questions.
> 1. Thank you for your attentive reading. Positive and negative concepts can only overlap at the level of the textual prompt, occurring simultaneously in a sentence. A concept cannot be both positive and negative at the same time. Building upon this premise, the objective of Refusal is to retain positive concepts while removing negative ones. One plausible approach to achieve this is by designing a distribution that is inversely proportional to the concept itself (as shown in Equation (3)). This would result in placing high likelihoods on positive concepts and low likelihoods on negative concepts, aligning with the goal of concept negation.
>
>     To illustrate, consider a text prompt like "mickey mouse eating an ice cream" and a pre-defined negative concept "N = {mickey mouse}". The remaining part of the prompt becomes the positive concept "P = {eating an ice cream}". In this scenario, a successful generative result would include the ice cream but exclude mickey mouse. While personalized generation that aims at producing specific contents is not our focus, a non-mickey mouse outcome or a missing object is acceptable. Our primary goal is to prevent the generation of negative content and ensure the proper generation of the positive concepts from the given prompt. However, the discussion of what content should replace the removed negative concepts is beyond the scope of this paper.
>
> For weakness:
> We will make revisions to the descriptions of Figures 1, 3, and 5 to enhance clarity. Additionally, Figure 5 will include more illustrative notes for better understanding.

---

> > ### Comment · Reviewer_2mKf · 2023-08-18
> > **rebuttal response**
> >
> > The subtlety of X but not Y has not been addressed, even by discussion without resolution, leaving the reader with the impression that this is a non-issue when imho it is.
> >
> > The authors seem only to consider the scenario where Y is actually named in X, but in this case why not just remove the words from X? As I said before if this approach type of approach will need to balance realizing X against not realizing Y.

---

> > > ### Author Response · Authors · 2023-08-18
> > > **Further Clarification**
> > >
> > > Thanks for the question. To clarify:
> > >
> > > In this work, we examine two scenarios involving the explicit mention of Y within X and its absence from X (as depicted in Figure 4, middle and right). Concerning the former scenario, rephrasing X to remove Y presents itself as an intuitive solution. Nonetheless, this approach encounters two challenges:
> > > 1. Creating a comprehensive list of prohibited terms encompassing abstract concepts like violent gore, specific art styles, etc., may pose difficulties (as illustrated in Figure 4, right);
> > > 2. It inadequately addresses the latter situation in which Y is not overtly present in X but emerges serendipitously in the generated output due to the pre-training knowledge of the diffusion model (as shown in Figure 4, middle). Our proposed method demonstrates its efficacy in effectively navigating these intricate circumstances.
> > >
> > > In essence, our method aims to achieve a harmonious state of "X without Y" within the attention map. This involves extracting Y through the cross-attention module, subsequently excluding it from the feature space, while preserving the residual features in an unaltered state. While this achieved equilibrium proves effective in the majority of scenarios, when confronted with more intricate situations (such as Y representing a substantial object or serving as an image background), a more refined equilibrium must be sought (as encapsulated in the Limitation section).
> > >
> > > Hopefully that addresses your question. Please feel free to ask if anything remains unclear.

---

### Official Review · Reviewer_YV3y · 2023-07-23

**Soundness:** 4 excellent
**Presentation:** 4 excellent
**Contribution:** 4 excellent
**Rating:** 7
**Confidence:** 3

**Summary:**

This paper proposes ProtoRe as a method for getting generative image models to refuse to show a certain concept. It incorporates the CLIP model’s knowledge, developing a prototype, and then refusing to generate (in effect) that prototype. The paper then presents a number of benchmarks showing ProtoRe’s performance both in generating the intended output and avoiding concepts that are supposed to be avoided.


**Strengths:**


The literature review is thorough, with other techniques described in detail.

The topic is important and timely and would be of general interest to the NeurIPS community.

The paper is clearly written overall.

The approach is interesting.



**Weaknesses:**

The paper focuses on easily recognizable concepts. Would be good to know if it generalizes to less easily recognizable concepts.

Quantitative performance seems strong, but it would be helpful to know how the qualitative evaluations (e.g., Figure 4) were selected.

-l260: Repetition of saying Fig 4 shows qualitative results



**Questions:**

1. The ImageNet subset chosen is 10 classes that are easily recognizable. Are the methods robust to using less easily recognizable concepts?

2. Figure 4 shows a number of qualitative refusals. How were these chosen and to what extent were they cherrypicked?



**Limitations:**

1. The ImageNet subset chosen is 10 classes that are easily recognizable. Are the methods robust to using less easily recognizable concepts?

2. Figure 4 shows a number of qualitative refusals. How were these chosen and to what extent were they cherrypicked?

---

> ### Author Rebuttal · Authors · 2023-08-07
>
> Thank you for your valuable feedback! Below we address your questions and concerns. Please feel free to post additional comments if you have further questions.
> 1. In response to less easily recognizable concepts, we divide them into three groups: small objects, large objects (including backgrounds), and abstract concepts. Our method demonstrates strong performance in handling small objects and abstract concepts, as illustrated in Figure 4 (b) and (c), respectively. However, due to the absence of pre-trained classifiers, quantitative results are not available. On the other hand, our method is comparatively less effective in dealing with large objects, such as a church. This limitation can be attributed to the relatively small refinement step size employed in Equation (7).
> 2. Figure 4 showcases six refusal examples in columns. The text prompts used are randomly chosen from the internet. Columns 1, 2, 3, 5, and 6 do not require deliberate selection. Additional results under various random seeds will be provided in the Appendix. In column 4, the text prompt is "A dog is playing in a park", and the refusal concept is "ball". Since the term "ball" does not explicitly appear in the text prompt, most of the generated images conditioned on this prompt do not contain the refusal concept. We carefully select the images that do include the concept of "ball" and examine their refusal results. This evaluation of implicit concept refusals helps prevent their accidental appearance in images when using regular prompts.

---

### Official Review · Reviewer_bbwA · 2023-07-24

**Soundness:** 3 good
**Presentation:** 2 fair
**Contribution:** 3 good
**Rating:** 6
**Confidence:** 3

**Summary:**

This paper introduces an innovative concept negation approach designed to effectively eliminate violent, unethical, or copyright-infringing content from generated images in diffusion-based models. The method operates during inference and stands apart from existing techniques by grounding concept negation in natural language. This unique characteristic enables defenders to easily specify which concepts should be considered harmless or offensive. To achieve this, the proposed method utilizes CLIP to generate the prototype of negative concepts and identifies the corresponding features associated with these concepts. These identified features are then utilized to refine the attention maps, purifying the content of negative concepts in the feature space. Extensive evaluation of the proposed method has been conducted across multiple benchmarks, demonstrating its superior performance compared to state-of-the-art methods. Notably, the proposed approach exhibits enhanced purification effectiveness and significantly improves the fidelity of generative images.



**Strengths:**

(1) In this academic paper, the authors introduce a novel "Prototype, Retrieve, and Refine" refusal method, designed to effectively eliminate violent, unethical, or copyright-infringing content from generated images during inference.

(2) Notably, the proposed method enables the definition of negative concepts using natural language, making it easily understandable for humans to specify which concepts should be considered objectionable.

(3) Extensive experimentation on multiple benchmarks showcases the superiority of the proposed PROTORE over existing approaches in terms of purification effectiveness and the fidelity of generated images across various settings.



**Weaknesses:**

(1) The academic paper introduces a method with promising potential; however, it would greatly benefit from stronger theoretical underpinnings. While the authors offer some intuitive explanations, they fall short of providing fully convincing arguments to support their approach.

(2) The architectural design presented in the paper appears solid, but it lacks sufficient detail, especially concerning the operational aspects of the "Prototype, Retrieve, and Refine" process. More comprehensive elucidation of these three steps would enhance the paper's clarity.

(3) An observed limitation of the proposed method lies in its performance with regard to removing large objects, such as backgrounds, from the generated images. Further investigation and improvement in this aspect would be valuable for its practical applicability.

(4) The creation of prototypes for negative concepts might be restricted in capturing all possible textual prompts related to these concepts.


**Questions:**

(1) In Equation (1), the mean of Gaussian distribution is $x_{t} + \sum_{i} \epsilon_{\theta}^{i}(x_{t}, t)$. However, the authors mentioned that they followed the work of “Compositional Visual Generation with Composable Diffusion Models”, where the mean is defined as $x_{t} - \sum_{i} \epsilon_{\theta}^{i}(x_{t}, t)$. The $\epsilon_{\theta}^{i}$ could be either the score function or the denoising direction in the context of DDPM. These two values differ by a negative sign. Could you explain this?

(2) Can you explain the clustering process in detail as Equation (4)?

(3) I don't quite understand the example of “a Mickey Mouse is eating ice cream”. If “Mickey Mouse” is considered a negative concept, filtering it out completely would result in undesired images. Otherwise, is it possible to generate another cartoon character eating ice cream? If yes, it still raises copyright issues. How did you deal with this issue?

(4) At each time step $t$ of the proposed algorithm, the value $A_t$ is computed using a pre-trained diffusion model as $A_{t}=\psi (z_{t}, \textbf{\textit{c}}, t)$. Does $A_t$ use to denote the output features produced by cross-attention operations, like the U-net architecture used by Stable Diffusion? After that, the algorithm feeds the modified attention map $A_{t}^*$ to the diffusion model, which generates the final denoised latent representation $z_{t-1}^*$. It seems that the above steps are different from the sampling process used in traditional diffusion models. Could you please provide more explanation about this?

(5) Does the proposed method slow down or speed up the sampling of the diffusion model? Could you report the computational complexity of sampling process?

(6) Can you provide some examples produced by the proposed method on the I2P dataset?


**Limitations:**

(1) The creation of prototypes for negative concepts might be restricted in capturing all possible textual prompts related to these concepts. Addressing this limitation and devising a more comprehensive approach to prototype generation would bolster the method's efficacy and accuracy.

(2) It seems that the proposed method performs worse in removing large objects (such as background) in the generated images.

---

> ### Author Rebuttal · Authors · 2023-08-08
>
> Thank you for your valuable feedback! Below we address your questions and concerns. Please feel free to post additional comments if you have further questions.
>
> (1) In this context, the $\epsilon_\theta^i$ signifies the output of a noise predictor implemented by a neural network. The plus or minus signs, regardless of their usage, only serve to signify changes in the noise distribution. For consistency with previous work, we revise the plus sign to a minus sign.
>
> (2) We employ three distinct clustering methods to derive the prototype prompts. In the case of single-label to single-class refusals (e.g., ImageNet), we utilize the embedding of the corresponding label as the prototype prompt. For multiple-labels to multiple-class refusals (e.g., I2P datasets), the clustering center is computed using K-means, which then serves as the prototype prompt. As for multiple dependent concepts refusals (as depicted in Figure 4), we combine all the concepts using commas or the word "and" to form the prototype prompt.
>
> (3) Given a text prompt such as "mickey mouse eating an ice cream" and a pre-defined negative concept represented by "N = {mickey mouse}", the remaining part of the prompt becomes the positive concept denoted by "P = {eating an ice cream}". In a desirable generative outcome, the result would contain the ice cream but exclude mickey mouse. While personalized generation targeting specific content is not our focus, the generation of a non-mickey mouse result or even an incomplete object is acceptable. Our primary objective is to prevent the generation of negative content and ensure the proper generation of positive concepts from the given prompt. However, the discussion of what content should be used to replace the removed negative concepts falls beyond the scope of this paper.
>
> Generating another cartoon character may still raise copyright issues. To tackle this problem, our approach can efficiently remove multiple concepts simultaneously (as demonstrated in Figure 4, left). Service deployers have the option to list multiple copyrighted characters, ensuring they do not appear in the generated image.
>
> An additional noteworthy benefit of this approach is its adaptability to the expiration dates of copyright on cartoon characters. Deployers can flexibly adjust the removal of characters (adding or deleting) in accordance with policies and regulations at any given time.
>
> (4) $A_t$ represents the output features generated through cross-attention operations, serving as an intermediate result of submodules within the U-net used by Stable Diffusion. Our approach does not alter the sampling rule for the diffusion model (e.g., the denoising process). Instead, we introduce modifications to $A_t$ specifically intended to remove certain concepts present within it. As these modifications are solely applied in the cross-attention module, our method can be viewed as a "restricted" conditional generation approach.
>
> (5) Our method has little effect on the sampling of diffusion models. Specifically, we introduced an extra ATTENTION operation with dimensions ranging from (2, 64, 320) to (2, 4096, 320) in the 16 CROSS ATTENTION modules of the diffusion model. With GPU acceleration, the computational overhead of constant-order attention is negligible. We compared the time overhead of generating 100 images before and after applying our approach on single RTX 3090:
>
> |  user prompt   | negative concept  | whole inference time | except for loading model |
> | :----: | :----: | :----: | :----: |
> |  a basket of fruit  |  | 551.3771 (s) | 535.2297 (s) |
> | a basket of fruit  | grape | 566.4718 (s)  | 554.9790 (s) |
> |
>
> (6) As per your suggestions, we will include additional experimental results on the I2P dataset in the appendix.

---

> ### Comment · Reviewer_bbwA · 2023-08-18
> **Thank you for your responses**
>
> I would like to express my gratitude for your responses addressing my inquiries. I will make the necessary adjustments to my evaluations based on the responses you've provided.

---

### Decision · Program_Chairs · 2023-09-21

**Decision:**

Accept (poster)

**Comment:**

I will recommend this paper for acceptance.

Overall the reviewers appreciated several aspects of the work: human-understandability of the work/approach (bbwA), the extensive experimentation (bbwA), and the authors' thorough literature review (YV3y). They found the paper interesting (YV3y, p72Z), the approach easy to use in practice (e3Lc), and felt that the results are strong (2mKf, e3Lc, WXn9, P7ZQ stated “this approach outperforms the baselines such as Stable Diffusion, composable diffusion model, and safe latent diffusion (with a few exceptions)”).

Reviewers did find several areas where they suggested improvement or asked questions: Reviewer bbwA noted that some of the theoretical underpinnings are weak and that some components of the work were not fully convincing (bbwA). Multiple reviewers noted that some concepts are probably not covered by this approach (bbwA, YV3y, e3Lc), and I would recommend more discussion of this be included in the next version. Other reviewers also had the following questions which could be good to address: “what should the system should be outputting when positive and negative prompts are overlapping?” (2mKf), "is it bounded by CLIP?" (e3Lc).

Engagement in Rebuttal: Engagement in rebuttal was active; bbwA and WXn9 raised their scores as a result. Authors took care to write an especially detailed response to a reviewer who expressed confusion and lack of expertise (WXn9), who appreciated this effort.

AC has several recommendations for authors: include more architectural design details (specified by bbwA), discussion of limitations associated with removing large objects (bbwA), computational complexity should be reported in the next version (bbwA), more info needed on qualitative eval and other analysis (YV3y, p7ZQ), dual use should be discussed somewhere (2mKf), the crawled negative prompts should be released, if possible/safe to do so (e3Lc)